# Identification of Key Modules and Genes Associated with Major Depressive Disorder in Adolescents

**DOI:** 10.3390/genes13030464

**Published:** 2022-03-05

**Authors:** Bao Zhao, Qingyue Fan, Jintong Liu, Aihua Yin, Pingping Wang, Wenxin Zhang

**Affiliations:** 1Department of Psychology, Shandong Normal University, Jinan 250358, China; zhaobao@sdnu.edu.cn (B.Z.); fanqingyue01@163.com (Q.F.); 2Shandong Mental Health Center, Shandong University, Jinan 250012, China; jintong@sdu.edu.cn (J.L.); yaihuade@163.com (A.Y.); 3Department of Life Science, Shandong Normal University, Jinan 250358, China; pingping.wang@sdnu.edu.cn

**Keywords:** major depressive disorder, gene expression, adolescent, WGCNA

## Abstract

Major depressive disorder (MDD) is a leading cause of disability worldwide. Adolescence is a crucial period for the occurrence and development of depression. There are essential distinctions between adolescent and adult depression patients, and the etiology of depressive disorder is unclear. The interactions of multiple genes in a co-expression network are likely to be involved in the physiopathology of MDD. In the present study, RNA-Seq data of mRNA were acquired from the peripheral blood of MDD in adolescents and healthy control (HC) subjects. Co-expression modules were constructed via weighted gene co-expression network analysis (WGCNA) to investigate the relationships between the underlying modules and MDD in adolescents. In the combined MDD and HC groups, the dynamic tree cutting method was utilized to assign genes to modules through hierarchical clustering. Moreover, functional enrichment analysis was conducted on those co-expression genes from interested modules. The results showed that eight modules were constructed by WGCNA. The blue module was significantly associated with MDD after multiple comparison adjustment. Several Gene Ontology (GO) terms and Kyoto Encyclopedia of Genes and Genomes (KEGG) pathways associated with stress and inflammation were identified in this module, including histone methylation, apoptosis, NF-kappa β signaling pathway, and TNF signaling pathway. Five genes related to inflammation, immunity, and the nervous system were identified as hub genes: *CNTNAP3*, *IL1**RAP*, *MEGF9*, *UBE2W*, and *UBE2D1*. All of these findings supported that MDD was associated with stress, inflammation, and immune responses, helping us to obtain a better understanding of the internal molecular mechanism and to explore biomarkers for the diagnosis or treatment of depression in adolescents.

## 1. Introduction

Major depression disorder (MDD), a serious and common mental disease, is characterized by a poor mood, cognitive impairment, social withdrawal, and even suicide tendencies [1]. It is one of the major causes of disability worldwide, with profound social and economic consequences [2]. Adolescence is a crucial period for the occurrence and development of depression [3], as depression often begins in adolescence [4], and among them has dramatically high incidence and prevalence [5,6]. There are obvious differences in the clinical symptoms [7,8], neurophysiological characteristics [9,10], and treatment responses [11] of adolescent and adult depression. Adolescent depression symptoms generally persist into adulthood [12], accompanied by health problems [13] and severe psychosocial deficits [14].

Although it is widely acknowledged that depressive disorder is a complicated disease affected by interactions between genetic and environmental factors [15], the underlying mechanisms remain unclear. Additionally, the current diagnostic systems based on self-reporting and clinical behavioral observation [16] do not adequately reflect the neurobiological alterations that induce the modified behavioral patterns in depressive patients [17]. Biomarkers of etiological pathways could provide objective, laboratory-based information to enhance the diagnosis of depression [18]. Genome-wide association studies (GWAS) have identified numerous single nucleotide polymorphisms (SNPs) related to major depression, and have found that depression-risk SNPs affect major depression susceptibility by changing gene expression in a tissue-specific manner. Some studies found SNPs that regulated the expression of genes in brain tissue which play putative roles in biological pathways related to synaptic signaling and neuronal system, and other studies revealed changed immune pathways in whole blood [19,20]. Regardless, in most instances, these depression risk loci are primarily located in noncoding regions of the genome, which are not immediately indicative of a causal gene, severely restricting biological interpretation [20]. Plenty of studies have used microarrays and RNA-seq to detect the gene expression profile of peripheral blood in depression, albeit the results have been inconsistent [21,22]. The transcripts discovered in prior studies were exhibited relations of various biological processes with depression, including nucleotide binding, chromatin assembly [23], DNA damage, DNA replication, repair processes, cell proliferation [24], neuronal apoptosis [25], inflammation [26], immune activation [24,25], signal transduction pathways [27], etc. The complexity of understanding the pathology of depression and discovering biomarkers for diagnosis and prognosis is attributed, at least in part, to the heterogeneity of the disease [28]. Besides, the limitations of traditional transcriptome approaches that identify significant single gene effects, such as over dispersion and multiple hypothesis testing [21,24], have hindered the discovery of genetic biomarkers for depression. Gene–gene interaction and network methods may summarize the variable spaces and enrich the information to further explain the specific genes and biological pathways underlying depression [29].

Modularity is a universal characteristic of biological systems [30,31]. A module is a collection of genes that are closely interrelated, and as a result, tend to share a biological function [32,33]. The dilemma of multiple hypothesis testing with RNA-Seq data is greatly alleviated by modular analyses of co-expression networks to confirm sets of differentially expressed genes in psychiatric disorders [34,35,36], including autistic spectrum disorder [37] and depressive disorder [38]. Thus, modular analysis has been proven to be a valuable method for studying the molecular underpinnings of complex diseases [39]. However, there have been few studies investigating the expression profile from a modular angle in depression so far. To our knowledge, no existing studies have examined the co-expression network modules of MDD in adolescents.

Weighted gene co-expression network analysis (WGCNA) is a systems biology approach for statistical analysis to investigate the complicated interactions between genes and clinical traits. It can be used to cluster genes that are highly correlated and construct gene co-expression network modules based on similar gene expression patterns, and to detect intramodular hub genes and related gene modules with clinical traits [40]. Co-expression modules are determined by unsigned hierarchical clustering [41], which is widely applied in multidimensional spaces [42]. The dynamic tree cut method is performed to identify the clustering dendrogram corresponding to modules [43] that are biologically meaningful [44]. WGCNA is commonly used to identify candidate biomarkers for prognosis and treatment in a variety of diseases, including cancer [45,46]. However, the application of WGCNA has received less attention for psychiatric illnesses, with only a few studies on depression. Zhao and colleagues determined enrichment of the estrogen signaling pathway and glucagon signaling pathway in brains of depression patients [47]. Another case-control study identified enrichment of apoptosis and B cell receptor signaling pathways associated with depression using the transcriptome from peripheral blood mononuclear cells [33]. Gerring et al. integrated the gene co-expression data with SNPs genotype data, discovering the enrichment of synaptic signaling, neuronal development, and cell transport pathways in co-expression modules in peripheral blood and numerous brain tissues related to depression [20]. These results suggested that gene expression affected major depression in a tissue-specific manner apart from the heterogeneity of MDD [48].

Herein, we attempted to identify gene co-expression network modules and hub genes in relation to MDD in adolescent via WGCNA technology. RNA-Seq data from peripheral blood mRNA were used to construct co-expression modules for the combined MDD and healthy control (HC) adolescent participants. The dynamic tree cut from the WGCNA tool was used to generate hierarchical clusters of similar size [40]. Gene Ontology (GO) and Kyoto Encyclopedia of Genes and Genomes (KEGG) enrichment analyses were then used to annotate the genes in the interesting modules regarding functionality and pathways. The present study may contribute to further understanding of the molecular mechanisms underlying the physiopathology of MDD adolescents [49,50].

## 2. Materials and Methods

### 2.1. Subjects

The adolescent MDD patients were recruited from the Shandong Mental Health Center, Jinan, China. Ten MDD Chinese adolescents (4 males, 6 females) and ten healthy controls, 14–19 years of age, were included as the subjects. Written informed assent and consent to participate in the study were obtained from all the participants.

MDD participants were included if they were experiencing their first onset of MDD according to the diagnostic criteria for depression of the DSM-V [51] and had a 17-item Hamilton Depression Rating Scale (HDRS) score ≥ 24. Clinical diagnoses were confirmed by two experienced psychiatrists applying the Structured Clinical Interview for DSM-V disorders [52]. Healthy controls were included if they did not meet the criteria for depression and scored ≤ 4 on the Patient Health Questionnaire-9 (PHQ-9) [18]. Patients and healthy controls matched each other by age, race, sex, and Tanner stage. Tanner Self-Rating Schematic Drawings were used to measure the pubertal stage [53]. The PHQ-9 was applied because it is specific to depression, with its items mapping onto diagnostic criteria for MDD [18].

Participants were excluded if they: (1) met diagnostic criteria for other psychiatric disorders (schizophrenia, bipolar disorder, etc.) or depression of organic etiology (e.g., hypothyroidism); (2) reported alcohol or substance abuse; (3) met diagnostic criteria for major medical illnesses or organic brain disorders; (4) were pregnant; (5) had received any antidepressant pharmacotherapy or individual psychotherapy before; (6) had a history of depression or other psychiatric disorders in themselves or family members.

### 2.2. Methods

#### 2.2.1. RNA-Seq Data Generation and Processing

Peripheral blood samples were collected into PAXgene Blood RNA tubues (Qiagen) and whole-blood RNA was extracted utilizing the PAXgene Blood RNA Kit (Qiagen) according to the manufacturer’s protocol. The RNA sample quality was assessed using the Agilent Bioanalyzer 2100 system (Agilent Technologies, Palo Alto, CA, USA). Following the manufacturer’s instructions, sequencing libraries were created. The paired-end RNA library was sequenced on an Illumina HiSeq 4000 sequencer (2 × 150 bp reading length). The generated sequence reads were trimmed and mapped to the human reference genome using Tophat2. Gene expression was normalized to the fragments per kilobase of transcript per million fragments mapped (FPKM). Cufflinks was employed to analyze gene expression and changes.

#### 2.2.2. Weighted Co-Expression Network Construction

The WGCNA algorithm, which was assumed to follow a scale-free distribution, was applied to perform the co-expression network module constriction of MDD [40]. The WGCNA package within R software was applied in this study. First, we employed correlation analysis to assess the co-expression relationship, and screened results for power values using the gradient method. The power values could facilitate calculating the dissimilarity coefficients of different nodes. Then, we utilized the topological overlap matrix (TOM) to build a hierarchical clustering tree, and we conducted approximate scale-free correlation analysis, which has been widely utilized for various diseases [54]. Furthermore, we prepared modules by assembling genes with high similarity into the same modules and genes with low similarity into separate modules. To ensure high dependability of the result, the minimum number of genes was set to 30.

#### 2.2.3. Module–Trait Correlation and Key Module Identification

Module–trait correlations were tested by applying the association between each module eigengene and trait, making it simple to determine the expression set (module) highly associated with the phenotype. Gene significance (GS) and module membership (MM) were defined for genes in the interested module eigengene (ME) [54]. Pearson correlation analysis is widely used as the default measure among co-expression network tools. In this study, Pearson correlation analysis between module–trait genes and phenotypes was used to calculate module-trait correlations to identify module members. The modules with adjusted *p*-values < 0.01 that showed the highest correlation coefficients were identified as the target modules. Furthermore, since the number of genes in each module was unequal, and some modules had large numbers of genes, the concept of dimensionality reduction was used to filter ME. Using ME to represent a huge body of genes for correlation analysis made gene module-to-module analysis easier [55].

#### 2.2.4. Functional Enrichment Analysis of Genes in Key Modules

Co-expressed genes are commonly involved in the same biological processes (BP) [56]. In a sense, modules are highly enriched with genes that share functional annotations [57]. Specific gene classes should be noticeably enriched with ideal modules [58]. GO enrichment analysis was carried out to provide further biological insight into the target module [59]. To explore which pathways were implicated in the strongly associated key modules, the pathway enrichment of this module was tested using KEGG pathway analysis [60]. KEGG can deduce high-level functions and utilities from molecular-level data generated by transcriptome sequencing [61]. *p*-values < 0.05 were defined as significant differences [62]. The top 10 records were extracted if there were more than 10 records.

#### 2.2.5. Hub Genes Identification

Hub genes were identified by two approaches. According to MM and GS, values of the genes in key modules were used to identify the hub genes. |GS| > 0.5, |MM| > 0.7 were employed as the thresholds to screen candidates in interesting modules. Furthermore, we conducted module analysis using cythubba with the Maximal Que Centrality (MCC) method in Cytoscape (Version 3.7.2) [63]. The top ten genes ranked by MCC values were considered as hub candidates [64,65,66,67]. The overlapped candidates obtained from the two methods were identified as the hub genes.

## 3. Results

### 3.1. RNA-Seq Data and Analysis

According to the t-test (Table 1), there were no differences in age, sex, or race between the MDD and HC groups. MDD and HC participants were matched for age, gender, and race. The MDD group displayed significantly more severe depressive symptoms than the HC group (based on HDRS score). The transcriptome data were collected using RNA-seq on the 20 subjects. After removing adaptor and low-quality reads, a total of 148.32 GB of clean data were produced by RNA-seq. At least 6.66 GB of clean data with >89.08 percent of them above Q30 was generated per sample (Appendix A). These results indicated that our transcriptome sequencing data were of high quality and sufficient for identifying differentially expressed genes (DEGs) and building the co-expression network. A total of 18,930 DEGs were discovered utilizing the criteria of a false discovery rate (FDR) < 0.05.

### 3.2. Identification of Key Modules

We constructed co-expression modules utilizing the R WGCNA package base on the expression values of 18,930 genes obtained from 20 subjects. The results revealed that all samples could be used in our analysis without the outliers from the hierarchical clustering tree. The power value was an essential parameter that influenced the independence and average connectivity of the network [40]. For the network topology, we used a soft-threshold power of four to exhibit the scale independence and average connectivity degree of the co-expression modules (Figure 1A). Then, the gene clustering tree of the co-expression network was used for module cutting after screening the power values. Eight modules decorated with diacritical colors were constructed (Figure 1B). The number of genes in each module varied from 35 to 295 (Table 2, Appendix A). The genes whose FPKM values < 5 in each sample were excluded in this analysis. The interactions of eight co-expression modules are depicted in Figure 1C. The co-expression relationship of genes in the key module was revealed to be significant by graphing the adjacency heatmap, and the eight modules were comparatively independent of each other (Figure 2A).

### 3.3. Identification of Key Co-Expression Network Modules for MDD

The module-trait correlation and GS values were calculated to identify the co-expression modules. The correlations between modules and traits were visualized using an eigengene dendrogram (Figure 2A) and a heatmap (Figure 2B). The interrelationships between co-expression modules and specific traits were determined by utilizing correlations between module eigengenes and traits. By calculating the correlation coefficient, the blue module was found to be the most closely related to MDD. The blue module unveiled significant correlations with MDD (r = 0.63, adjusted *p*-value = 0.003), indicating that genes involved in this module most likely participate in the occurrence and development of MDD in adolescents. Gender, age, and Tanner stage were not shown to be associated with gene modules. There were 215 genes in the blue module. Scatterplots of GS and MM were plotted in the blue module (Figure 2C).

### 3.4. Functional Enrichment Analysis of Genes in the Blue Module

Further analysis of the blue module, which included 215 genes, was performed to explore more valuable genes. GO and KEGG enrichment analyses were conducted to detect gene functions and biological pathways closely related to MDD in the blue module. The results exhibited that the genes of the blue module mainly participated in regulation of transcription via RNA polymerase II promoters in response to hypoxia, positive regulation of histone H3-K9 methylation, and calcium ion transport in BP ontology. In cellular component (CC) ontology, genes were primarily involved in the actin cytoskeleton, and are related to postsynaptic density and neuron projection terminus. Genes in the blue module were involved in inward rectifier potassium channel activity, phosphatidylserine binding, positive regulation of T cell mediated immunity, etc., in molecular function (MF) ontology (Figure 3A, Appendix A).

The results of KEGG pathway enrichment analysis of the DEGs between the MDD and HC groups in the blue module are shown in Figure 3B (Appendix A), which illustrated that genes in the blue module are mainly involved in apoptosis, TNF signaling, and NF-kappa β signaling, and they are related to ubiquitin mediated proteolysis.

### 3.5. Gene Interactions within Blue Module and Hub Genes Identification

The constructed gene interaction network can be imported into the Cytoscape software for visualization [63]. The hub candidate genes were screened based on GS and MM values of genes in the blue module. As a result, 33 hub candidate genes were obtained from blue module (Figure 4, Appendix A). Additionally, using the MCC algorithm in Cytoscape, the top ten were screened as hub candidates (Appendix A). Combining the two methods, at last, five down-regulated genes were identified as hub genes closely related to MDD in adolescents: *CNTNAP3*, *IL1RAP*, *MEGF9*, *UBE2W*, and *UBE2D1*.

## 4. Discussion

MDD, a serious and common mental disorder, has sparked broad concern and emerged as a major public health problem. In the present study, RNA-seq was performed on peripheral blood samples of ten MDD adolescents and ten healthy control adolescents, yielding 18,930 differentially expressed genes. Applying the WGCNA method, eight co-expression modules were constructed for these DEGs to explore the relationship between MDD and the transcriptome. Only one co-expression module, the blue module, was shown to be strongly linked to MDD. GO enrichment analysis illustrated that the genes in the blue module were predominantly involved in positive regulation of histone methylation, positive regulation of T cell mediated immunity, and postsynaptic density. The DEGs in the blue module are involved in apoptosis, the TNF signaling pathway, and the NF-kappa β signaling pathway, which are cellular functions and pathways associated with the response to stress and immunity. Five down-regulated transcripts, including *MEGF9*, *IL1RAP*, *CNTNAP3*, *UBE2W*, and *UBE2D1*, were identified as the hub genes of MDD in adolescents. In prior gene expression studies of MDD, a single gene was commonly used as the unit of analysis for differential expression. There have been few studies focused on co-expression gene modules via transcriptome analysis. To our knowledge, this is the first research to identify co-expression modules of MDD in adolescents.

WGCNA is concerned with the association between co-expression network modules and complex diseases (in this case, MDD) [68]. Module-based analysis can be utilized to determine gene modules that are remarkably related to depression without overfitting in the lower-dimensional hypothesis space. In this study, the WGCNA algorithm provided more effective information which had the potential to explore biomarkers for the diagnosis or treatment of depression in adolescents.

The present study identified one co-expression module correlated with major depression. GO and KEGG enrichment analysis elucidated that the genes in the blue module are predominantly involved in positive regulation of T cell mediated immunity, postsynaptic density, apoptosis, TNF signaling, and NF-kappa β signaling. The enrichment levels of genes in postsynaptic density, apoptosis, and signaling pathways were consistent with some results of GWAS and transcriptome studies, showing that depression risk genes are involved in central nervous system development, synaptic plasticity, and immune pathways [19,20]. Moreover, existing studies of co-expression modules conducted by WGCNA identified enrichment of apoptosis, B cell receptor signaling in blood [33], estrogen signaling, glucagon signaling in the brain [47], synaptic signaling, neuronal development, and cell transport pathways in peripheral blood and numerous brain tissues [20]. These results suggested some overlap between brain and peripheral blood; meanwhile, gene expression affected major depression in a tissue-specific manner apart from the complexity and heterogeneity of MDD [48].

It has been recognized that environmental stress, such as childhood maltreatment, emotional trauma, and interpersonal conflict, are closely associated with depression [69,70,71]. However, the underlying mechanisms are largely unknown. Mounting evidence has manifested strong correlations among epigenetic changes and depression [69,70,72]. Histone acetylation is a category of common epigenetic modification that functions in the regulation of DNA-templated reactions, such as transcription. In animal models, chronic stress increased anxiety-like behavior, accompanied by the alteration of histone acetylation in the forebrain, which showed that rats which suffered early life stress displayed greater stress vulnerability [73,74]. The histone methylation processes have also been linked to a variety of psychiatric diseases [75]. The present study revealed that the blue module was primarily associated with the positive regulation of histone H3-K9 and H3-K4 methylation. The results demonstrated the correlation between stress-related epigenetic alteration and depression.

According to the inflammatory and neurodegenerative hypothesis [76], depression is closely correlated with inflammation and immune function, accompanied by neurodegeneration and declined neurogenesis [77]. A body of evidence have demonstrated the features of major depressive disorder, including the interrelationships between the inflammatory response and cell-mediated immune activation [78]. The activation of the inflammatory response system during stress exposure is associated with hypothalamic–pituitary–adrenal axis (HPA) hyperactivity, indicating that HPA hyperactivity in depression is contributed by proinflammatory cytokines, such as IL-1β and TNFα [77]. Consequently, cell-mediated immune activation contributes to the serotonergic disturbances in depression. Moreover, the activation of NF-κB and other inflammatory pathways aroused from stress stimuli, triggered neuronal apoptosis [78]. Our results exhibited that genes in the blue module were significantly involved in regulation of T cell-mediated immunity and predominantly associated with pathways linked to apoptosis, TNF signaling, and NF-kappa B signaling; and IL1RAP, associated with inflammation [79], was detected to be down-regulated in MDD in this research. Evidence at the molecular level above illustrated the correlation between inflammation and depression.

*UBE2W* and *UBE2D1* relevant to immune and tumors are involved in the ubiquitin-proteasome system, which mediates protein degradation in eukaryotes [80]. Previous studies have found that *UBE2W* expression is significantly correlated with the immune environment, and it is associated with neurodegenerative diseases and Huntington’s disease [81,82]. Although previous studies have not identified *UBE2W* and *UBE2D1* to be closely correlated with depression, transcriptome analyses have discovered that depression is linked to a range of biological processes which may be involved in inflammation and immune activation. In this research, *UBE2W* and *UBE2D1* were detected to be down-regulated in MDD adolescents, indicating down-regulation the in immune function of major depression. The results were consistent with some previous GWAS studies which showed altered immune pathways in peripheral blood [21,83]. Gene expression studies found that genes of major depression heavily involve the immune response against infections as well [84]. These findings implied that depression might influence the immune response, and immune system regulation should be applied to MDD therapy in adolescents.

*CNTNAP3*, a member of the CASPR (contactin associated protein) family, has been found to be associated with several psychiatric disorders in prior studies. An animal experiment found that *CNTNAP3* was detected in various regions of the mouse brain (e.g., cortex, frontal lobe, corpus callosum, hippocampus) and was suggested to play a role in cellular recognition of neural networks [85]. The dysregulation of *CNTNAP3* was found to be linked to neurocircuit impairment in schizophrenia patients’ brains [86,87]. Researchers found that *CNTNAP3* deficiency leads to delayed motor learning [88]. However, the relationship between *CNTNAP3* and depression has rarely been studied. The expression of *CNTNAP3* was down-regulated in depressed patients in the current study, demonstrating the weaker impairment in neurocircuit. The results indicated that the dysregulation of *CNTNAP3* might affect brain function and consequently influence the onset of depression. Therefore, *CNTNAP3* might be validated as a candidate gene for transcript biomarker in further studies.

Notably, this study found *MEGF9* to be a novel candidate critical gene associated with adolescents. MEGF9 is a transmembrane protein that contains several epidermal growth factor-like repeats. *MEGF9* is recognized to have a role in the development, maintenance, and injury response of the nervous system [40]. Prior studies illustrated that *MEGF9* was highly expressed in neuronal and glial cells of the central nervous system and peripheral nervous system [49]. In this study, the dysregulation of *MEGF9* expression in MDD patients indicated that the development of the nervous system and injuries to it were closely correlated with MDD in adolescents. The dysregulated expression of BP associated with development might be the underlying mechanism of the dramatic increase in the incidence and prevalence of depression in adolescence. Therefore, *MEGF9* might have the potential to be the specific biomarker for MDD in adolescents.

Although the pathophysiological mechanisms of MDD in adolescents are unclear, mounting evidence has revealed that genetic factors [89] and environmental factors such as stress are dominantly implicated in the pathology of depression [90]. The results of the current study revealed genes associated with stress (e.g., positive regulation of histone methylation), inflammation (e.g., *IL1RAP*), and immune response (e.g., *UBE2W* and *UBE2D1*), supporting the hypothesis that the pathological processes of depression are regulated by stress-related genes and immunological inflammation [91]. Individuals with depressive disorder showed an increased inflammatory response to stress, accompanied by elevated inflammatory genes. Moreover, environmental factors such as stress could influence the occurrence and development of depression by affecting signal transduction pathways, possibly due to the fact that genes involved in signal transduction mechanisms are more sensitive to the emotional trauma experienced by individuals [92,93,94].

To the best of our knowledge, the current study was one of the early attempts to explore the transcriptome in MDD adolescents by WGCNA and identified key genes as potential biomarkers. In particular, we focused on transcriptome analysis of MDD in adolescence, which is a critical period in the development of major depression that has received too little attention. However, several limitations should be considered. Firstly, the sample size was relatively small, and would be worthwhile to obtain sufficient data from a larger sample size in further study. Secondly, the WGCNA strategy summarized the score for a module to one value, which lead to putative information loss at the single gene level [40]. Moreover, the results in this study were based on WGCNA data mining without further validation by experiments. Finally, another potential limitation of this study was adopting gene expression from peripheral blood. Although peripheral blood is an easily accessible source of cells, supplying some new insights for clinical biomarkers of depression [95], it includes various cell types and may not detect brain-specific mechanisms [21,22]. Multi-tissue gene expression approaches may have the potential to elucidate the complex biological mechanisms underlying major depressive disorder.

## 5. Conclusions

In the current research, WGCNA was applied to construct a gene-weighted co-expression network to detect key gene modules and hub genes which were highly related to MDD in adolescents. The blue module was identified, and genes in the blue module were mainly enriched with positive regulation of histone methylation, positive regulation of T cell mediated immunity, and postsynaptic density by GO analysis. Moreover, we found that apoptosis, TNF signaling, and NF-kappa B signaling were involved in MDD. Five hub genes related to immunological inflammation and the nervous system with significant changes in depressed adolescents were identified as candidate biomarkers for MDD in adolescents. Therefore, the current study supported the inflammatory and neurodegenerative hypothesis of depression [71]. All of these findings reinforced that MDD is associated with stress, inflammation, and immune responses, helping us to better understand the internal molecular mechanisms and investigate biomarkers for the diagnosis or treatment of depression in adolescents.

## Figures and Tables

**Figure 1 genes-13-00464-f001:**
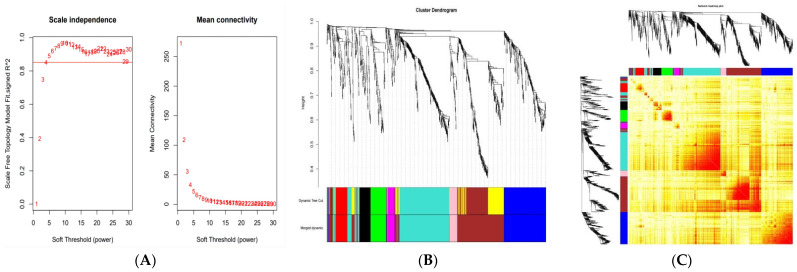
(**A**) Clustering of samples and determination of soft thresholding power. (**B**) Hierarchical dendrogram of genes. Eight modules were constructed, each decorated with a different color. (**C**) Visualizing gene co-expression network TOM. A light color represents a low overlap, and gradually darker red indicates high overlap.

**Figure 2 genes-13-00464-f002:**
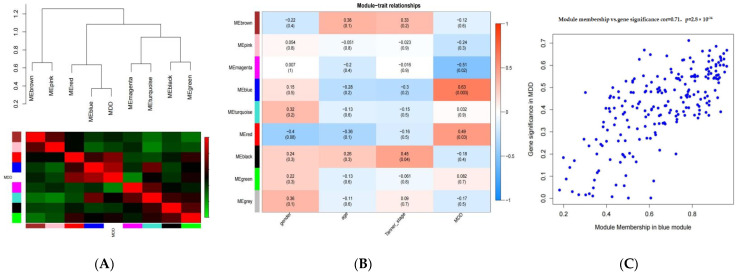
(**A**) The eigengene adjacency heatmap. The dendrogram indicates that the blue module is highly related to MDD. (**B**) Correlation heatmap of modules and traits. The adjusted *p*-value and correlation coefficient are shown in each cell. (**C**) Scatterplot of Gene significance (GS) for depression VS.module membership (MM) in the blue module.

**Figure 3 genes-13-00464-f003:**
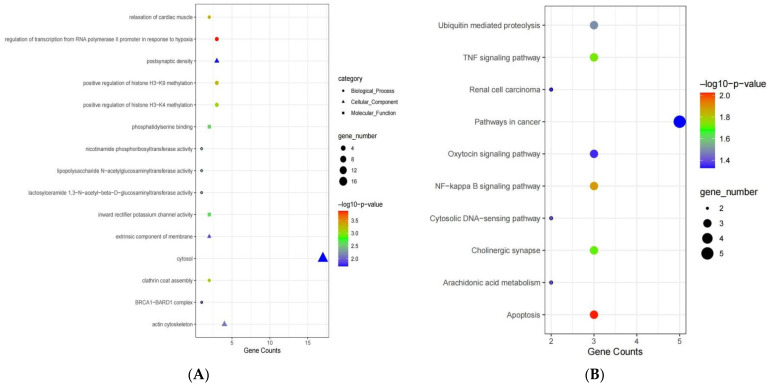
(**A**) GO enrichment analysis of genes in the blue module. (**B**) KEGG pathway analysis of genes in the blue module.

**Figure 4 genes-13-00464-f004:**
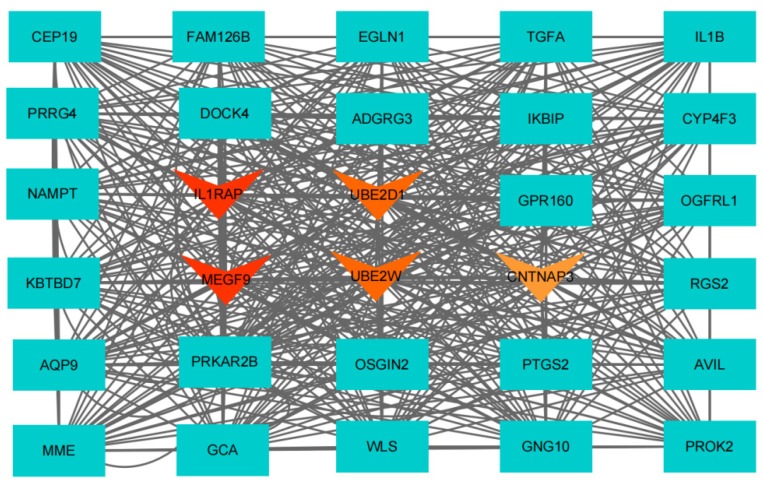
Networks of hub genes in the blue module. Triangular nodes represent hub genes; downward nodes represent downregulated genes, with darker red representing higher scores.

**Table 1 genes-13-00464-t001:** Characteristics of MDD and HC adolescents.

Variable	HC (*n* = 10)	MDD (*n* = 10)
Age	16.5(1.6)	15.9(1.4)
Gender (male/female)	4/6	4/6
HDRS Mean (s.d)	1.5(0.7)	42.1(11.8)
PHQ-9 Mean (s.d)	1.4(1.3)	23.3(1.8)

**Table 2 genes-13-00464-t002:** Numbers of genes in the eight modules.

Module Colors	Gene Number
black	56
blue	215
brown	295
green	77
magenta	35
pink	41
red	58
turquoise	294

## Data Availability

The datasets for this manuscript are not publicly available due to data confidentiality. Requests to access the datasets should be directed to Wenxin Zhang, zhangwenxin@sdnu.edu.cn.

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
