# Peer review of "Identification of Key Modules and Genes Associated with Major Depressive Disorder in Adolescents"

_genes, 2022, doi:10.3390/genes13030464_

Round 1

Reviewer 1 Report

Zhao et al performed a gene co-expression analysis of major depression using expression data from peripheral blood of adolescents. WGCNA was used to identify network modules in case and control samples. The authors identified a single module significantly associated with depression and enriched with biological pathways related to the immune system. These results align with earlier findings of blood gene expression in major depression (e.g. Jansen et al 2016), supporting the immune dysfunction hypothesis. The network-based approach has been performed in previous papers, however this study was performed in adolescents instead of adults. . Please address the follow comments:

General comments

    The authors combined case and control data for the gene co-expression networks. The network structure is likely biased by case sample, leading to inappropriate biological conclusions. Have the authors tried to construct modules from case and control data separately?
    The introduction needs to mention previous work on co-expression networks underlying major depression (e.g. Gerring et al 2019 [PLoS Gen]). How does your work build on these published results?
    There needs to be greater discussion of the limitations of your study (e.g. small sample size, use of peripheral blood for gene expression)
    Greater discussion of genetic data for major depression required. How do these results align with results from genome-wide association studies? Why are there differences in enriched pathways? Could the gene expression results reflect environmental effects (e.g. stress)?

Specific comments:

    Methods, line 157: were P values corrected for multiple testing?
    Results: How were the P values adjusted in figure 3B?
    Results, line 170-182: information on file size is not required
    Results, figure 4: these data could be presented differently, the current interaction map tell us nothing about the connectivity or importance of each gene in the network.

Author Response

Response to Reviewer 1 Comments

Point 1: The authors combined case and control data for the gene co-expression networks. The network structure is likely biased by case sample, leading to inappropriate biological conclusions. Have the authors tried to construct modules from case and control data separately?

Response 1: Thank you very much for your insightful comment. We do realize there are some coexpression studies using WGCNA in cases (Gerring et al., 2019), and some studies combine case and control data for the gene co-expression networks (Le et al., 2018; Zhang, 2020; Wittenberg et al., 2020). For instance, Gerring and colleagues built gene co-expression networks individually for 13 brain tissues and whole blood in major depression cases (Gerring et al. , 2019), and other researchers applied case-control design to conduct modular analysis of comparing differentially expressed genes (DEGs) (Le et al., 2018; Zhang et al., 2020). These two strategies helped to explore the underlying molecular mechanisms of major depression from different perspectives, which were widely used by depression researchers. However, it is difficult to apply both strategies in the same study. In this paper, we explored a case-control study to detect co-expression modules of DEGs. In the future, we will consider exploring modular analyses in cases, as the reviewer suggested, to further analyze the data and obtain a multidimensional understanding of the molecular mechanisms of major depression.

References

Gerring, Z.F., Gamazon, E.R.; Derks, E.M. A gene co-expression network-based analysis of multiple brain tissues reveals novel genes and molecular pathways underlying major depression. PLOS Genet 2019, 15, e1008245.

Le, T.T., Jonathan, S.; Hideo, S. Identification and replication of RNA-Seq gene network modules associated with depression severity. Translational Psychiatry 2018, 8(1),180.

Zhang, G.; Xu, S.; Yuan, Z. Weighted gene coexpression network analysis identifies specific modules and hub genes related to major depression. Neuropsychiatric Disease and Treatment 2020, 16, 703-713.

Wittenberg, G.M.; Greene, J.; Vértes, P.E.; Drevets, W.C.; Bullmore, E.T. Major depressive disorder is associated with differential expression of innate immune and neutrophil-related gene networks in peripheral blood: A quantitative review of whole-genome transcriptional data from case-control studies. Biol Psychiatry 2020 88(8), 625-637.

Point 2: The introduction needs to mention previous work on co-expression networks underlying major depression (e.g. Gerring et al. 2019 [PLoS Gen]). How does your work build on these published results?

Response 2: Thanks for your valuable comments. Based on your suggestions, we added more information about previous work on co-expression networks underlying major depression in the introduction, which demonstrating that our study based on the method and results of some previous studies.

As you can see on page 2 line 92-95 and page3 line 96-103 “ However, the application of WGCNA has been received less attention on psychiatric illnesses, with only a few studies on depression. Zhao and colleagues determined enrichment of estrogen signaling pathway and glucagon signaling pathway in brains of depression patients. Another case-control study enrichment of apoptosis and B cell receptor signaling pathways associated with depression using the transcriptome from peripheral blood mononuclear cells (Le et al., 2018). Gerring et al. integrated the gene co-expression data with SNP genotype data, discovering the enrichment of synaptic signaling, neuronal development and cell transport pathways in co-expression modules in peripheral blood and numerous brain tissues related to depression (Gerring et al., 2019). These results suggested that gene expression affected major depression in a tissue-specific manner apart from the heterogeneity of MDD (Gerring, 2021).”

Reference

Le, T.T.; Jonathan, S.; Hideo, S. Identification and replication of RNA-Seq gene network modules associated with depression severity. Translational Psychiatry 2018, 8(1),180.

Point 3: There needs to be greater discussion of the limitations of your study (e.g. small sample size, use of peripheral blood for gene expression)

Response 3: Your insights deepen our thinking about the results. In the discussion, we added limitations of our study as suggested.

As you can see on page 10 line 400-415 : “To the best of our knowledge, the current study was one of the early attempted to explore the transcriptome in MDD adolescents by WGCNA and identified key genes as potential biomarkers. In particular, we focused on transcriptome analysis of MDD in adolescence, which is a critical period in the development of major depression with lack of attention that deserves further study. However, several limitations should be considered. Firstly, the sample size is relatively small, and it is worthwhile to obtain sufficient data from a larger sample size in further study. Secondly, the WGCNA strategy summarized the score for a module to one value, which lead to putative information loss at the single gene level (Langfelder & Horvath, 2008). Moreover, the results in this study are based on WGCNA data mining without further validation by experiments. Finally, another potential limitation of this study is to adopt the gene expression from peripheral blood. Although peripheral blood is an easily accessible source of cells supplying some new insights for the clinical biomarker of depression (Zhang et al., 2020), it includes various cell types and may not detect brain-specific mechanisms (Mostafavi et al.,2014; Jansen,2016). Multi-tissue gene expression approaches have the potential to elucidate the complex biological mechanisms underlying major depressive disorder.”

References

Langfelder P.; Horvath S. WGCNA: an R package for weighted correlation network analysis. BMC Bioinf  2008, 9, 559.

Zhang, G.; Xu, S.; Yuan, Z. Weighted gene coexpression network analysis identifies specific modules and hub genes related to major depression. Neuropsychiatric Disease and Treatment 2020, 16, 703-713.

Mostafavi, S. et al. Type I interferon signaling genes in recurrent major depression: increased expression detected by whole-blood RNA sequencing. Mol. Psychiatry 2014, 19, 1267–1274.

Jansen, R.Gene expression in major depressive disorder. Mol. Psychiatry 2016, 21, 444.

.

Point 4: Greater discussion of genetic data for major depression required. How do these results align with results from genome-wide association studies? Why are there differences in enriched pathways? Could the gene expression results reflect environmental effects (e.g. stress)?

Response 4: Thanks for the thoughtful comments. First, we agree that greater discussion of genetic data for major depression should be added in this paper. According to the reviewer’s suggestion, we add some information and discussion about genome-wide association studies in the introduction and discussion section.

Specifically, 1) the second paragraph in the part of Introduction (Please see page 2 line 49-58): “Genome-wide association studies (GWAS) have identified numerous single nucleotide polymorphisms (SNPs) that related to major depression, and have found that depression risk SNPs affect major depression susceptibility by changing gene expression in a tissue-specific manner. Some studies found SNPs that regulated the expression of genes in brain tissue which played putative roles in biological pathways related to synaptic signaling and neuronal system, while other studies revealed changed immune pathways in whole blood (Zhou et al., 2020; Gerringet et al., 2019). Nevertheless, in most instances, these depression risk loci primarily located in noncoding regions of the genome, which are not immediately indicative of a causal gene, severely restricting biological interpretation (Gerring et al. , 2019). “

  • the last paragraph in the part of Discussion (please see page 8 line 303-311; page 9 line 312-317): “The present study identified one co-expression module correlated to major depression. GO and KEGG enrichment analysis elucidated that the genes in the blue module were predominantly enriched in positive regulation of T cell mediated immunity, postsynaptic density, apoptosis, TNF signaling pathway and NF-kappa B signaling pathway. The enrichment of genes in postsynaptic density, apoptosis and signaling pathways were consistent with some results of GWAS and transcriptome researches, which showing that depression risk genes involved in central nervous system development, synaptic plasticity, and immune pathways (Zhou et al., 2020; Gerring et al., 2019). Moreover, existing studies of co-expression modules conducted by WGCNA identified enrichment of apoptosis, B cell receptor signaling in blood (Le, 2018), estrogen signaling pathway, glucagon signaling pathway in brains(Zhao, 2020), synaptic signaling, neuronal development, and cell transport pathways in peripheral blood and numerous brain tissues (Gerring et al., 2019).”

Second, as concerned with the differences in enriched pathways, the complexity and heterogeneity of MDD, and the tissue-specific gene expression manner might affected differences in enriched pathways of major depression(Please see page 9 line 314-317).

The revision is as following: “ These results suggested some overlap between brain and peripheral blood, meanwhile gene expression affected major depression in a tissue-specific manner apart from the complexity and heterogeneity of MDD (Gerring, 2021).”

Third, given the lack of environment factors involved in the study, gene expression results of the present study did not reflect the environmental effects directly. However, some genes (e.g. IL1RAP) and enriched pathways (e.g. TNF signaling pathway, histone acetylation and T cell mediated immunity) identified in the study have been shown to be related to stress in previous studies (The Network and Pathway Analysis Subgroup of the Psychiatric Genomics Consortium, 2015). Additionally, it has been recognized that environmental stress, such as childhood maltreatment, emotional trauma and interpersonal conflict, are closely associated with depression (Heim et al., 2008; Slavich et al., 2014; Park et al., 2019). Therefore, we suggested that these gene expressions may be a stress-induced alteration of gene expression, which in turn affected depression. For instance,the enrichment of TNF signaling pathway and NF-kappa B signaling pathway and the dysregulation of the five hub genes. We have added explanation of this part in this revision.

As indicated on page 8 line 288-292 : “ The DEGs in module blue were enriched in apoptosis, TNF signaling pathway and NF-kappa B signaling pathway which were cellular functions and pathways associated with the response to stress and immunity.”

page 9 line 318-330: “ It has been recognized that environmental stress, such as childhood maltreatment, emotional trauma and interpersonal conflict, are closely associated with depression (Heim et al., 2008; Slavich et al., 2014; Park et al., 2019). Mounting evidence have manifested strong correlation among epigenetic changes and depression (Heim et al., 2008; Slavich et al., 2014; Turecki et al., 2016). Histone acetylation is a category of common epigenetic modification that functions in the regulation of DNA-templated reactions, such as transcription. In animal models, chronic stress increased anxiety-like behavior, accompanied by the alteration of histone acetylation in the forebrain, which showed that rats suffered early life stress displayed greater stress vulnerability (Jakobsson et al., 2008; Kyoung et al., 2016). The histones methylation processes have also been linked to a variety of psychiatric diseases (The Network and Pathway Analysis Subgroup of the Psychiatric Genomics Consortium, 2015). The present study revealed that the blue module were primarily enriched in positive regulation of histone H3-K9 and H3-K4 methylation terms. The results demonstrated the correlation between stress-related epigenetic alteration and depression.”

References

Zhou, D.; Jiang, Y.; Zhong, X.; Cox, N. J.; Liu, C.; Gamazon, E. R. A unified framework for joint-tissue transcriptome-wide association and Mendelian randomization analysis. Nat Genet 2020, 52, 1239– 1246

Le, T. T.; Jonathan, S.; Hideo, S. Identification and replication of RNA-Seq gene network modules associated with depression severity. Translational Psychiatry 2018, 8(1),180.

Zhao, Y.; Wang, L.; Wu,Y.J.; Lu, Z.Q.; Zhang, S.Y. Genome-wide study of key genes and scoring system as potential noninvasive biomarkers for detection of suicide behavior in major depression disorder. Bioengineered 202011, 1189-1196

Gerring, Z. F.; Gamazon, E. R.; Derks, E. M. A gene co-expression network-based analysis of multiple brain tissues reveals novel genes and molecular pathways underlying major depression. PLOS Genet. 2019, 15, e1008245.

Gerring, Z. F . Dissecting genetically regulated gene expression in major depression. Biological Psychiatry 2021, 89, 6.

Heim, C.; Newport, D.J.; Mletzko, T.; Miller, A.H.; Nemeroff, C.B. The link between childhood trauma and depression: insights from hpa axis studies in humans. Psychoneuroendocrinology 200833(6), 693-710.

Slavich, G.M.; Irwin, M.R. From stress to inflammation and major depressive disorder: A social signal transduction theory of depression. Psychological Bulletin 2014, 140(3), 774–815.

Park, C.; Rosenblat, J.; Brietzke, E. Stress, Epigenetics and Depression: A systematic review. J Neuroscience & Biobehavioral Reviews. 2019.

Turecki, G.; Meaney, M.J. Effects of the social environment and stress on glucocorticoid receptor gene methylation: A systematic review. Biological Psychiatry 2016, 79(2), 87–96.

Jakobsson, J.; Cordero, M.I.; Bisaz, R. KAP1-mediated epigenetic repression in the forebrain modulates behavioral vulnerability to stress. Neuron. 2008, 60(5), 818-831.

Kyoung, S.M.; Ngoc, L.N.; Hong, L.C. Early life stress increases stress vulnerability through BDNF gene epigenetic changes in the rat hippocampus. Neuropharmacology 2016, 105, 388-397.

The Network and Pathway Analysis Subgroup of the Psychiatric Genomics Consortium. Correction: Corrigendum: Psychiatric genome-wide association study analyses implicate neuronal, immune and histone pathways. Nat Neurosci. 201518, 926 .

Point 5: Methods, line 157: were P values corrected for multiple testing?

Response 5: Thank you very much for your comment. We corrected the wrongly uploaded data in Table S4. The corrected Table S4 data did not present the adjusted p-value because they were not significant after multiple testing. According to previous studies (Zhang et al., 2020; Li et al., 2021; Li et al.,2020; Chen et al., 2019; Chen et al., 2020), P-values <0.05 were defined as significant differences in the current study (Benjamini et al., 2001). Although the uncorrected significance is not very rigorous, we report it in this manuscript with some value for understanding the underlying mechanisms of major depression.

References

Zhang, G.; Xu, S.; Yuan, Z.; Shen, L. Weighted gene coexpression network analysis identifies specific modules and hub genes related to major depression. Neuropsychiatr Dis Treat 2020, 16, 703-713.

Li, J.X.; Cao, X.J.; Huang, Y.Y.; Li, Y.P. Investigation of hub gene associated with the infection of Staphylococcus aureus via weighted gene co-expression network analysis. BMC Microbiol 2021, 21(1), 329. 

Li, X.; Wang, C.; Zhang, X, et al. Weighted gene co-expression network analysis revealed key biomarkers associated with the diagnosis of hypertrophic cardiomyopathy. Hereditas 2020, 157,1.

Chen, R.; Ge, T.; Jiang, W.; Huo, J.; Chang, Q.; Geng, J.; Shan, Q. Identification of biomarkers correlated with hypertrophic cardiomyopathy with co-expression analysis. J Cell Physiol 2019, 234, 12.

Chen, X.; Wang, J.; Peng, X et al. Comprehensive analysis of biomarkers for prostate cancer based on weighted gene co-expression network analysis. Medicine (Baltimore) 2020, 99, 14.

Benjamini, Y.; Drai, D.; Elmer, G.; Kafkafi, N.; Golani, I. Controlling the false discovery rate in behavior genetics research. Behav. Brain Res 2001, 125, 279–284.

Point 6: Results: How were the P values adjusted in figure 3B?

Response 6: Thanks for your valuable comment. The p-values showed in Figure 3B which were matched with Table S4 were not adjusted because they were not significant after Benjamini–Hochberg correction. Although unadjusted significance is not very rigorous, these results are of some value for exploring the underlying biological mechanisms of depression.

References

Zhang, G.; Xu, S.; Yuan, Z.; Shen, L. Weighted gene coexpression network analysis identifies specific modules and hub genes related to major depression. Neuropsychiatr Dis Treat 2020, 16, 703-713.

Li, J.X.; Cao, X.J.; Huang, Y.Y.; Li, Y.P. Investigation of hub gene associated with the infection of Staphylococcus aureus via weighted gene co-expression network analysis. BMC Microbiol 2021, 21(1), 329. 

Li, X.; Wang, C.; Zhang, X, et al. Weighted gene co-expression network analysis revealed key biomarkers associated with the diagnosis of hypertrophic cardiomyopathy. Hereditas 2020, 157, 1.

Chen, R.; Ge, T.; Jiang, W.; Huo, J.; Chang, Q.; Geng, J.; Shan, Q. Identification of biomarkers correlated with hypertrophic cardiomyopathy with co-expression analysis. J Cell Physiol 2019, 234, 12.

Chen, X.; Wang, J.; Peng, X et al. Comprehensive analysis of biomarkers for prostate cancer based on weighted gene co-expression network analysis. Medicine (Baltimore) 2020, 99, 14.

Point 7: Results, line 170-182: information on file size is not required

Response 7: Thanks for your comment. We have deleted the information on file size, and revised this part in this revision as suggested.

As you can see on page 5 line 200-203 : “After removing adaptor and low-quality reads, a total of 148.32 GB of clean data was produced by RNA-seq. At least 6.66 GB of clean data with >89.08 percent of them above Q30 was generated per sample (Supplementary Table S1). ”

Point 8:   Results, figure 4: these data could be presented differently, the current interaction map tell us nothing about the connectivity or importance of each gene in the network.

Response 8: Thanks for your comment. Following the reviewer’s advice, we have revised Figure 4 in Results section. We hope that this revision shows the connectivity or importance of each gene in the network more clearly.

The revised figure is as following: (Please see page 8 line 275-277)

Figure 4. Networks of hub genes in module blue. Triangular nodes represent hub genes; down nodes represent downregulated genes, with darker red representing higher scores.

Reviewer 2 Report

In this interesting study, Zhao and collaborators identified key modules and genes associated with MDD in adolescents. The study is well written and scientifically sound. The are some points of concern described below.

1- The authors stated that this study followed ethical standards, could they clarify whether it was approved by an ethical committee and provide an approval number/reference?

2- Was the RNA-seq performed in-house (Illumina HiSeq 4000 sequencer)? Is there any identification number that could be written in the methods section?

3- Is the RNA-seq data deposited into any database repository?

4- Did the authors assess/observe any gender differences?

5- Figure 2A is mirrored.

6- Could the authors describe, either in topic 3.5 or in discussion session, whether the reported genes are up- or down-regulated in MDD patients?

7- It would be interesting to see how the expression of the 5 identified genes correlate with the respective HDRS score for each patient (e.g. higher score, lower expression).

Author Response

Response to Reviewer 2 Comments

Point 1:  The authors stated that this study followed ethical standards, could they clarify whether it was approved by an ethical committee and provide an approval number/reference?

Response 1: Our study was approved by the Ethics Committee of Shandong Normal University. The approval reference is attached.

Point 2:  Was the RNA-seq performed in-house (Illumina HiSeq 4000 sequencer)? Is there any identification number that could be written in the methods section?

Response 2: Yes, the RNA-seq was performed in-house. Peripheral blood samples were collected into PAXgene Blood RNA tubes (Qiagen) according to the manufacturer’s protocol and then stored to −80℃ until use. According to the protocol, currently available data shows stabilization of cellular RNA for at least 96 months at –20℃ or –70℃. After we obtained the blood samples from 20 subjects, the total RNAs were extracted for the subsequent study.  In order to express the sample collection method more clearly, we have made a corresponding modification in the manuscript but without identification number. In this study, 20 samples were collected over 19 months and the sequencing was conducted on an Illumina HiSeq 4000 sequencer at the same time.

As you can see on page 3 line 138-140 ”Peripheral blood samples were collected into PAXgene Blood RNA tubues (Qiagen) and Whole-blood RNA was extracted utilizing the PAXgene Blood RNA Kit (Qiagen) according to the manufacturer’s protocol. ”

Point 3: Is the RNA-seq data deposited into any database repository?

Response 3: No, the datasets for this manuscript are not publicly available due to data confidentiality. Requests to access the datasets should be directed to Wenxin Zhang, zhangwenxin@sdnu.edu.cn. We have added the data availability statement to this revision (Please see page 11 line 446-448 ).

Point 4:  Did the authors assess/observe any gender differences?

Response 4: Thanks for your valuable comment. There have been some studies showing gender differences in MDD (Seney, 2018), and we are also interested in this. However, the limitation of our sample size made it non-significant to assess the gender difference. In addition, according to the existing literature, there is not much data on gender differences in module construction studies (Le et al., 2018; Zhang et al., 2015; Zhang et al., 2020). Therefore, in this study, we only did the exploration of sex difference analysis at transcription level without constructing the modules in males and females separately. In addition, we have analyzed shared DEGs in each of the male and female groups. The results showed differences in major histocompatibility complex (MHC) genes (e.g. HLA-DBQ1, HLA-DRA) between male and female adolescents. However, the results were not suitable for the same paper, thus they were not included in this manuscript.

References

Seney, M. L.; Zhiguang. Opposite molecular signatures of depression in men and women. Biological Psychiatry 2018, 84, 18-27.

Zhang, W.; Cao, Y.; Wang, M.; Ji, L,; Chen, L.; Deater-Deckard, K. The dopamine D2 receptor polymorphism (DRD2 TaqIA) interacts with maternal parenting in predicting early adolescent depressive symptoms: Evidence of differential susceptibility and age differences. J Youth Adolesc 2015, 44(7), 1428-40.

Le, T.T.; Jonathan, S.; Hideo, S. Identification and replication of RNA-Seq gene network modules associated with depression severity. Translational Psychiatry 2018, 8(1),180.

Zhang, G.; Xu, S.; Yuan, Z. Weighted gene coexpression network analysis identifies specific modules and hub genes related to major depression. Neuropsychiatric Disease and Treatment 2020, 16, 703-713.

Point 5:  Figure 2A is mirrored.

Response 5: Thanks for your comments. We are sorry for the mirrored figure, and we have made a correction to this part. (Please see page 6 line 240)

Point 6:  Could the authors describe, either in topic 3.5 or in discussion session, whether the reported genes are up- or down-regulated in MDD patients?

Response 6: Thanks for the insightful comments. The reported five hub genes are down-regulated in MDD patients. We have added explanation of down-regulated genes in this revision.

Specifically, 1) the third sentence of 3.5 (please see page 7 line 269-271) : “ Combining the two methods, at last, five down-regulated genes were identified as hub genes closely related to MDD in adolescents: CNTNAP3 (Contactin Associated protein-like 3), IL1RAP (interleukin- 1 receptor accessory protein), MEGF9 (Multiple EGF-like domains 9), UBE2W (ubiquitin-conjugating enzyme E2W) and UBE2D1 (ubiquitin-conjugating enzyme E2D1). ”

  • the first paragraph in the part of Discussion (please see page 8 line 291-292) : “Five down-regulated transcripts including MEGF9, IL1RAP, CNTNAP3, UBE2W and UBE2D1 were identified as the hub genes of MDD in adolescent.”

  • The fifth paragraph in the part of Discussion (please see page 9 line 326-327) :”The expression of CNTNAP3 was down-regulated in depressed patients in the current study, demonstrating the weaker impairment in neurocircuit.”

  • The sixth paragraph in the part of Discussion (please see page 9 line 356-362) : “In this research, UBE2W and UBE2D1 were detected to be down-regulated in MDD adolescents, indicating the down-regulation in immune function of major depression. The results were consistent with some previous GWAS studies which showed altered immune pathways in peripheral blood [21,84]. Gene expression studies found that genes of major depression dominantly enriched in immune response against infections as well [85]. These findings implied that depression might influence the immune response and the target of immune system regulation should be applied to MDD therapy in adolescents. ”

Point 7:  It would be interesting to see how the expression of the 5 identified genes correlate with the respective HDRS score for each patient (e.g. higher score, lower expression).

Response 7: Thanks for your comment. Following the reviewer’s suggestion, Pearson correlation analysis was conducted to test the correlation between 5 hub genes and depression score. However, the result showed no significant correlation between each gene and HDRS score (see Table 1 below).
